# Manufacturing and Characterization of Wide-Bundle Bamboo Scrimber: A Comparison with Other Engineered Bamboo Composites

**DOI:** 10.3390/ma15217518

**Published:** 2022-10-26

**Authors:** Yuan Hu, Luyao Xiong, Yanbo Li, Kate Semple, Vahid Nasir, Hugo Pineda, Mei He, Chunping Dai

**Affiliations:** 1Department of Wood Science, University of British Columbia, Vancouver, BC V6T 1Z4, Canada; 2Research Institute of Forest Products, Jiangxi Academy of Forestry, Nanchang 330013, China; 3Jiangxi Provincial Forest Resources Conservation Center, Nanchang 330038, China

**Keywords:** bamboo composites, wide-bundle bamboo scrimber, engineered bamboo products, fiberization, mechanical properties, dimensional stability

## Abstract

Controlling the variability in mat structure and properties in bamboo scrimber (BS) is key to producing the product for structural applications, and wide strip scrimber (WBS) is an effective approach. In this study, the effects of scrimmed bamboo bundle morphology and product density on the properties of WBS were investigated. WBS panels were manufactured and tested using wide (200 to 250 mm) bamboo strips with different fiberization intensity. Maximum strength properties (flexural, compressive, and shear strength), and lowest thickness swelling and water absorption were achieved with three or four passes due to the higher resin absorption by strips. For balanced product cost and performance, we recommend 1–2 fiberization passes and a panel density of 0.9–1.0 g/cm^3^. Panel mechanical properties were compared with other common bamboo composites. Bamboo scrimber products were highly variable in properties due to differing manufacturing processes, element treatments, and suboptimal mat structure. Products including laminated bamboo lumber and flattened bamboo made from nonfiberized elements show markedly different relationships between strength and elastic properties mostly due to inadequate bonding between the laminae, which causes premature bond-line failure. This study helped improve the understanding of the structure–property relationship of engineered bamboo products while providing insights into process optimization.

## 1. Introduction

Bamboo is one of the strongest, fastest-growing plants in the world. Bamboo forests are widely distributed across the subtropical regions of Asia, Africa, and Latin America [1]. According to the International Network of Bamboo and Rattan (INBAR), the total area of global bamboo forests was more than 35 million hectares in 2020 [2]. Bamboo has a shorter growth/harvest cycle than trees and can be harvested within 5 years. Bamboo products are extremely versatile and can be manufactured to high strength and stiffness for potential use in sustainable construction development [3].

In North America and Northern Europe, timber has been traditionally and widely used for construction [4,5]. As lumber prices increase, developing engineered bamboo materials may help supplement timber building materials. Engineered bamboo uses wood adhesives and hot pressing to convert round, hollow, and variable bamboo culms into dimension ‘lumber’ and panels for construction giving a wide range of properties suitable for different applications. Engineered bamboo is promising as a rapidly renewable, bio-based construction material that can also offer enhanced and uniform mechanical performance. Research and development of engineered bamboo are essential for regions in countries such as China and India with abundant bamboo but low timber resources. Engineered bamboo for building construction needs to be lower in density than some existing products such as scrimber (for fastening); more standardized, uniform, and predictable in density and properties; and lower in cost. Understanding the factors contributing to the high variability in bamboo composite properties and manufacturing products with better-controlled and standardized properties and performance is crucial for their acceptance as structural materials in the building and construction industry.

Engineered bamboo products can be generally classified into two groups: (1) products with laminated structures such as laminated bamboo lumber (LBL); and (2) strand and fiber-based products with more random their structure, such as bamboo oriented strand board (OSB) and scrimber. Most structural wood composites used in building framing are laminated, including plywood, laminated veneer lumber (LVL), cross-laminated timber, and glued-laminated timber. These are multilayer products with continuous glue lines, but failure to adhere to standardized manufacturing and quality control protocols such as ANSI PRG 320 (developed for softwood glulam and CLT) can lead to premature bond-line failure and reduced modulus of rupture (MOR) [6]. Strand and fiber-based products have a higher degree of randomness in their mat structures and discontinuous glue lines. This paper deals with a fiberized bamboo scrimber (BS) product called wide bundle scrimber or WBS, designed to fit the laminated structure category to overcome the limitations and deficiencies caused by random mat structure. 

BS adapts the crushing and flattening technology developed in Australia to convert small diameter pine thinnings to bundles of splinters that are coated in resin and hot-compressed back into solid composite lumber [7,8]. BS is made from crushed or ‘fiberized’ bamboo bundles soaked in water-soluble phenol-formaldehyde (PF) resin and compressed to billets or panels with the required density and thickness [9]. BS has a high utilization rate of raw bamboo, high density, and good physical and mechanical properties [10,11]. It also has a high PF resin solids content (15 to 17%) compared with most other wood and bamboo composites and is very heavily compacted at >10 MPa, giving it a tight grain texture suitable for high-wear flooring and decking applications as well as garden construction and certain civil engineering applications [12,13,14]. Approximately 60 bamboo scrimber manufacturing enterprises presently exist in China, with an annual production rate of 600,000 m^3^ [15]. The earliest recorded research on bamboo scrimber materials dates back to the 1980s [16]

The fabrication process consists of cold molding and hot curing, or, for panels, of a lengthy cool-in and cool-out pressing procedure. Products range in density from about 0.85 to 1.25 g/cm^3^, with high variability in mechanical properties [17,18,19]. Many studies have investigated in depth the mechanical properties [20,21,22,23], flame retardant properties [24], dying properties [25], anticorrosion and antimildew properties [26], and supplementary element heat treatments for bamboo scrimber [27]. Various processing equipment such as flattening and fiberization devices, cold-compress machines, special curing kilns, multilayer hot-presses with water cooling systems, and hot-press molds with movable stops have been developed to suit BS fabrication [28]. 

Bamboo scrimber can be made with high mechanical properties but at the expense of heavy processing—mechanical culm flatting, cracking requiring much resin usage to ‘repair’, heavy compaction, and lengthy press times to consolidate. The high densification leads to excessive thickness well. Conventional BS is made from narrow flattened bamboo strips which have had the inner and outer skins, or ‘cortex’, removed to improve impediments to resin bonding [29], but this also reduces the culm utilization rate. Narrow strips, randomized strip placement in mats, and lack of standardization of the manufacturing process result in an overly densified product with a wide range of properties which limits its application as a structural building material. Narrow strips do not align as well in the mat and create more random gaps and overlaps within each layer, or ‘laminate’, requiring excessive compaction to provide adequate contact pressure between elements.

Wide bundle scrimber (WBS) and stitched bundle scrimber or bamboo laminated veneer lumber (BLVL) are more recent and effective approaches to convert BS to more of a plywood or LVL-type laminated mat structure and therefore reduce the compaction required for adequate element contact. The objective of this study was to examine the effects of fiberization extent (number of passes through toothed rollers) on resin absorption by strips as well as associated panel density on mechanical properties including flexural strength, shear strength, and thickness swell. Modifications to the manufacturing process to enhance dimensional stability and mechanical properties within a suitable density range are discussed. The study also aims to contrast the properties of WBS with those of traditional BS and other engineered bamboo products to gain broader understanding of the processing-structure-property relationships of bamboo-based composites. 

## 2. Materials and Methods

### 2.1. Materials

Fresh 4–5-year-old Moso bamboo (*Phyllostachys pubescens*) culms were obtained from Yingtan city, Jiangxi province, China. The bamboo trees were 20 m in height with a culm diameter of 80 to 100 mm and wall thickness of 7 to 11 mm. The moisture content of the raw bamboo was 35% to 45%, and density was between 0.5 g/cm^3^ and 0.6 g/cm^3^. The harvested bamboo was delimbed and cut to 2-meter-long culms which were then split longitudinally into two to four sections for flattening and fiberization.

### 2.2. Experimental Design

Two factors were tested in the study: (Experiment a) extent of fiberization—four levels; and (Experiment b) board density—five levels, as shown in Table 1. Three replicate panels were made for each level. The different levels of board density were achieved by adjusting the weight of resin-soaked fiberized bamboo layers used in the mat and compressing them to the same thickness. The extent of fiberization for the board density variation was 3 passes through the rollers which were considered to be the optimum fiberization extent to balance resin usage with panel properties.

### 2.3. Fiberization

The culm halves were flattened and decorticated into fiberized bundles (Figure 1) measuring between 200 and 250 mm in width while in the fresh green state using a multi-purpose fiberization machine [30] and then cut into 460 mm segments. The fiberized strips were dried in an oven at 85–90 °C for 3 h to a moisture content of 6–7% and kept at room temperature for 1 week before testing and panel fabrication. During the fiberization process, a series of parallel cracks were formed in the culm wall which separates the wall tissue into a series of larger or smaller interconnected fiber bundles. The rollers have discontinuously dislocated convex teeth to generate cutting and peeling forces in the longitudinal direction of the bamboo wall. Linear and point cracks were formed longitudinally along the bamboo culm, while extrusion forces were applied to the radial section resulting in the loosening and stretching of the broomed strip in the transverse (across the grain) direction [31]. Several pieces of fiberized bamboo veneer were randomly selected, and the diameter of the thickest bundle of fibers in each piece was measured using vernier calipers. Information on bundle sizes and utilization rate of different groups is given in Table 2 and Table 3). Utilization rate was estimated from the mass of the culm sections before and after passing through the rollers. The green removal rate (also given in Table 3) was estimated from a randomly cut 5 cm × 5 cm specimen and imaged under a depth-of-field 3D super-deep-scene microscope. The lasso tool in Photoshop software was used to identify and measure the remaining green skin area seen in Figure 1.

### 2.4. Calculation of Specific Surface Area (SSA)

The concept of specific surface area was proposed here to quantify the extent to which the bamboo strips are fiberized. Pictures of the cross-section of the bundles were taken for each level of fiberization, and their average thicknesses were measured. The pictures were then cropped to enclose only the cross-section and then binarized to obtain black-and-white images using the image processing and analysis library available in MATLAB^®^ R2019b. The number of white pixels was counted and divided by the total pixel area to obtain the occupied portion. The white portion was also divided into non-touching regions, and the perimeter of each region was calculated. The value of SSA was calculated as the sum of the perimeters of all regions divided by the total occupied area. Since the units of the SSA results are 1/pixel, the thicknesses of each photographed bundle were measured and used as a scale to convert the results from pixels to mm. The average resolution of the pictures is 80 pixel/mm.

### 2.5. Panel Fabrication

The dried bundles were dipped in liquid phenolic resin for 3–4 min and drained for 4 min to remove excess resin for reuse. The phenolic resin was supplied by Beijing Taier Chemical Co., Ltd. (Beijing, China), with a solids content of 46.56%, viscosity of 42 CPs, and pH of 10–11. The resin dosage (of liquid resin in Table 3) was estimated by measuring the increase in mass. The resin-coated strips were then oven-dried to an MC of 10–12%. The resinated, dried strips were hand-laid into a mold with side pressure baffles with the inner face of the strips facing the core and the outer faces facing the surfaces for uniform symmetry in the vertical direction [32]. To adjust density, between 6 and 8 strips were used per panel, which resembled unidirectional LVL in configuration. The 460 mm × 200 mm mats were hot-pressed to a 20 mm target at 145 °C under 5 MPa compaction pressure for 30 min (1.5 min/mm hot press time) using a cold-in–cold-out (water-cooled) schedule. Note the compaction pressure is less than half that of traditional BS due to the uniform laminates and there being no gaps or overlaps between strips within the mat. The implications of this are discussed further in Section 3.4. Total panel pressing time was over an hour, accounting for precompaction and postcooling to 60 °C before removal. The laboratory press was supplied by Shanghai Wood-based Panel Machinery Factory Co., Ltd. (Shanghai, China). The panel was further cooled to ambient temperature then and conditioned before cutting for density and mechanical properties specimens.

### 2.6. Physical and Mechanical Properties Testing

The physical and mechanical properties of the WBS panels were tested according to Chinese national standards GB/T 30364 and GB/T 20241. For dimensional stability, the thickness swelling rate (TS), width swelling rate (WS), and water absorption rate (WA) after immersion in boiling water for 4 h was followed by oven drying at 63 °C for 20 h then reimmersion in boiling water for 4 h. Specimen size was 20 mm × 50 mm × 50 mm, with 12 specimens measured and averaged per group. Flexural strength, parallel-to-grain compressive strength, and shear strength were measured using a Jinan Meters microcomputer-controlled wood-based panel universal testing machine (MWD-w10). Test specimen size was 20 mm × 20 mm × 450 mm (MOR), 20 mm × 40 mm × 120 mm (SS), and 60 mm × 20 mm × 20 mm (CS). The average of 6 specimens was taken for each group. The internal morphology of each type of panel was observed in 10 mm × 10 mm blocks using a scanning electron microscope (SEM, S-3400, Hitachi, Tokyo, Japan) after sputter-coating with gold.

## 3. Results

### 3.1. Structural Characteristics of WBS

The difference between WBS strips and earlier iterations of BS scrimmed strips is shown in Table 2. WBS has much wider fiberized strips of up to 250 mm, whereas traditional and narrow bamboo scrimber strips are between 30 and 60 mm wide. Narrow BS is characterized by variability in planar panel density caused by poorly controlled mat lay-up which creates overlaps and gaps requiring very high compaction pressures to achieve adequate contact and bonding of elements [19,33,34,35]. WBS allows for more controlled mat lay-up—an intermediary between haphazardly-laid traditional BS and well-aligned bamboo bundle LVL (see Figure 2). The method also permits lower compaction pressures (~5 MPa) used here, leading to more uniform density and improved panel properties and performance. From BS factory production data, the production efficiency of WBS is about 5–6 times higher than that of traditional BS. The utilization rate of WBS was estimated to be 92% since the inner and outer cortex layers are not removed. The method leads to a 40% increase in the utilization rate compared to traditional BS.

The percentage weight of liquid resin absorbed by the WBS strips with different brooming passes is shown in Table 3. Liquid resin absorption after one pass was about 26.5% but increased to 72.4% with four passes due to the larger extent of cracks, higher specific surface area [33] (see estimates in Table 3), and greater damage and removal of inner and outer cortex. While the resin absorption increased 2.7 times, the value of SSA was more than doubled after four passes, with a majority of cracks occurring during the early rounds of brooming. More fiberization passes decrease the fiber bundle diameter and increase the SSA. Greater SSA also leads to higher-bonded surfaces and resin-infused mechanical interlocking networks. The average diameter of the fiber bundles after four passes through the fiberization rollers was 1.89 mm, giving an estimated contact surface area of about twice that of strips with one pass.

The fiberization extent must be balanced with the correct amount of resin uptake to ‘repair’ the cracks and provide adequate properties without becoming uneconomic in terms of resin consumption. Excessive fiberization requires greater energy consumption and cost and reduced utilization rate to balance the need to mechanically flatten and create supple, conformable strips with excessive fiberization. The mechanical properties’ results also indicate no further gains from more than three passes, suggesting that only up to two or at most three passes are needed.

### 3.2. Effect of Fiberization on the Scrimber Properties 

Strength properties for each fiberization level are shown in Figure 3, and thickness and width swelling and water absorption rate of WBS after 4 h and 28 h are shown in Figure 4.

Flexural properties (modulus of rupture, MOR, and modulus of elasticity, MOE) increased between one and three passes, with no further gain at four passes. Compression strength (CS) and shear strength (SS) showed a steady increase with the number of passes up to three, as this is governed by the interface strength between two laminated faces. More passes mean more fissures in the adherend surfaces and more resin in the panel, creating stronger and better resin-interlocking between adherends and stronger bond interfaces. However, more than three passes likely compromises the structural integrity of the bamboo tissue to the extent that it starts to degrade composite mechanical properties and become uneconomic in terms of resin usage required to ‘repair’ accompanied by loss of strength properties. Fiberization with few passes retains larger and solid bamboo tissue in the length direction, with the resin coating ensuring minimal loss of mechanical properties compared to unbroken tissue.

The dimensional stability indicators thickness swell (TS), width swell (WS), and water absorption (WA) were significantly reduced with fiberization frequency from one to three with a further small gain after four passes. The greater resin uptake and content in panels with more intensively fiberized strips (i.e., greater resin-to-bamboo ratio) explain the trend, as PF resin is water-resistant compared with the hydrophilic bamboo tissue. From the results, no more than two to three fiberization passes are needed for optimal mechanical and dimensional properties.

### 3.3. Effect of Density on the Scrimber Properties

Flexural properties MOR and MOE both increase with density, as shown in Figure 5. The greatest increase was from 0.9 g/cm^3^ to 1.1 g/cm^3^. Ideally, for minimizing processing costs and time, engineered bamboo should be not much greater in density than the bamboo parent tissue, and results from this study suggest high-recovery, resin-bonded BS products could be manufactured to 0.9–1.0 g/cm^3^ density. If mat layering is carefully controlled, as discussed in Figure 2, then the minimum required product densification can be reduced and still maintain adequate element contact and bonding [36]. 

The shear and compression strength (SS and CS) with board density are shown in Figure 6a,b. Unlike the flexural properties, the increase in SS and CS with panel density is linear. This is likely due to the fact that both shear and compressive strengths are more directly related to bonding strength, which in turn correlates to density through its impact on interelement contact development. On the other hand, bending MOE and MOR are more dependent on inter-element contact and bonding when the density is low. As the density increases, sufficient bonding develops, and the MOE and MOR will be governed by other factors such as fiber length and orientation.

The thickness swell at 4 and 28 h with panel density are shown in Figure 7. TS increased with density up to 1.1 g/cm^3^, then decreased as the ratio of water-resistant resin was high enough to hamper absorption and tissue swelling. The lowest 28 h TS was at 0.9 g/cm^3^ (6.5%), since the extent of cell compaction and internal stress is the lowest. 

Lower board density and higher void volume fraction allow easier penetration by water during soaking treatment. At a panel density of 0.8–1.0 g/cm^3^, there are still some open pores conducive to more rapid penetration and absorption of water. However, despite the increased liquid uptake, there is also less compacted tissue per unit volume of the panel to absorb moisture and swell, resulting in lower overall irreversible TS in the low-density panels. At the other end of the density range, the very high tissue compaction restricts moisture ingress and swelling during the limited moisture exposure time. Given unlimited water exposure time, the highest density panels would be expected to have the highest TS.

Board density increases with higher volumes of bamboo bundles pressed to the same thickness due to a higher compaction rate. During hot-pressing and material softening, the vessels mostly undergo buckling and collapse, as seen in Figure 8b,c, which occurs during the viscoelastic softening and tissue relaxation during hot-pressing. If compaction and deformation occur rapidly at the beginning of hot-pressing before the material has heated sufficiently to plasticize and absorb the stress, then localized microfractures can occur, weakening the bamboo parent tissue. Similarly, if compaction is too slow and the resin heats and precures, then bond integrity is compromised. The fundamentals of BS hot-pressing, which should be similar to the consolidation behavior of wood composites [37,38,39,40], are under investigation and will be published in a future paper.

After pressing, some internal stress remains within the panel. With exposure to moisture, the compressed cells absorb water via diffusion and restore their original shape causing swelling, delamination, and internal flaws. After continuous water boiling, elastic deformation continues to increase as internal stress is gradually released [41]. Rupture of internal bond lines may occur, caused by the heavy compaction, which exacerbates deformation and swelling, causing increased irreversible thickness swell rate. Such high-density and low-dimensional stability have in the past greatly limited the cost-effectiveness and commercial applications of BS. One reason for the high resin requirement and low cost-effectiveness of BS is to help contain TS caused by high compaction. Panels in this study had relatively low TS due to the high ratio of water-resistant resin coating the tissue and reduced free void space restricting water entry. 

One strategy for reducing moisture uptake and swelling without needing more resin is to adjust the resin formulation. For example, Zhu et al. [42] discussed the ability of low-molecular phenolic resin to provide more uniform and effective penetration into bamboo tissue, and it was believed to be the primary contributor to the dimensional stability of BS that was made at different density levels. Higher dimensional stability may result from small resin molecules entering, polymerizing, and occupying smaller voids and more of the hydrophilic functional groups in the tissue. More frequent deposits of low molecular weight resin on bamboo cell walls and their internal pits can effectively decrease the swelling properties of testing specimens when immersed in water. The deposit of polymers on the bamboo cell wall is the fundamental contributor to the improvement of the dimensional stability of the scrimber board.

### 3.4. Impact of Densification on the Microstructure of Scrimber 

Figure 8 compares the transverse section of raw bamboo (density = 0.65 g/cm^3^) and WBS at both ends of its density range of between 0.9 g/cm^3^ and 1.3 g/cm^3^ At the lower densification, localized deformations occurred in the parenchyma cells. As the densification increases, the deformation extends to vascular bundles, especially in the vessels as seen in the high-magnification image in Figure 8c. The reduction of large lumen space influences water absorption and TS. At excessive densification levels, cracks and debonding (Figure 8c) start to develop due to localized shear failures, particularly if the tissue has not sufficiently heated and softened during compaction and/or the resin has precured and become brittle. These failures do not necessarily weaken the global strength properties as observed in MOR and SS, but could negatively impact other structural properties such as fatigue and duration of load. 

Compaction reduces the between- and within-strip void fraction, creating more effective contact and adhesion of the resin. The strength of the bamboo fiber in the length direction and the bonding strength between fiber bundles largely determine the strength properties of the scrimber. At the same panel thickness, higher density is achieved by adding more layers, resulting in a higher degree of element compaction ratio and densification during hot pressing. At too low a density, the weak bonding cannot provide sufficient stress transfer between elements and hence become the weakest link in the composite. As density increases element surface contact and pressure, the bond quality increases to the point where it exceeds the strength of the bamboo tissue, and tissue strength becomes the limiting factor. High compaction forces more resin into smaller pores and voids in the tissue, creating better mechanical interlocking. High-density panels contain fewer voids, resulting in many more bond contact points for stress transfer between the elements. Properties must be balanced with increased panel weight and the time, energy, and resin required to press higher-density BS and the reduced dimensional stability that accompanies very high compaction of BS products. The results from this study with wide strips favor minimal passes through the rollers (1 to 2) to reduce tissue break-up and resin consumption, and densification to no more than 0.9 to 1 g/cm^3^ to achieve adequate strength properties for structural purposes. Wide strips are recommended for BS to help reduce the number of elements per mat, the problems associated with mat heterogeneity, and flaws caused by misalignment and random overlaps and gaps between narrow strips, as illustrated and discussed earlier in Figure 2. 

### 3.5. Performance Comparison of WBS with Other Engineered Bamboo Materials

To further understand the structure–property relationships of bamboo composites, the MOR and MOE of different engineered bamboo materials reported from past studies in the literature are compared using Ashby plots in Figure 9 and reported values in Table 4. Bamboo products may be classified into five product types: natural raw bamboo (RB), laminated bamboo lumber (LBL), flattened bamboo (FB) bamboo scrimber (BS and WBS), and bamboo bundle laminated veneer lumber (BLVL). Figure 9 shows mechanical properties are generally positively correlated with product density.

Raw bamboo (RB) is natural bamboo without any treatment or densification and therefore has baseline density and strength properties for the bamboo type. RB has lower strength properties than most engineered bamboo products, especially BS and BLVL, due to the densification and large resin addition to these products. Different species of raw bamboo exhibit a large range in density but relatively low correlation with strength properties.

Laminated bamboo lumber (LBL) has a low densification ratio, just enough to compress and bond the flat smooth strips together, and is most similar to that of RB, as it requires little densification for bonding [29]. LBL is made by stacking and gluing uniformly milled bamboo elements, producing a uniform element shape, size, and orientation in the product. The average strength of structural LBL is similar to flattened bamboo and hardwoods [62]. Several studies have demonstrated the unique properties of structural LBL [62,63,64,65,66]. Structural members made of LBL can be created in any cross-section and shape and are adaptable to different heights and spans of members. Of particular note is that while the MOE of LBL increased with density, the MOR did not show a significant increase with density because MOR was governed more by glue-line strength, which could fail prematurely, rather than transferring stress to the tissue. Unlike BS, which is a composite mixture of fiber bundles encased in large amounts of resin that can withstand loading even when its structure begins to fail, the stress transfer capability of LBL collapses, resulting in a masked density effect on bending strength. In contrast, the MOE of LBL increases with increasing density because the MOE reflects the material’s ability to resist elastic deformation, i.e., the linear portion gradient of the stress–strain curve before the glue line becomes affected.

Similarly, laminated FB has comparable properties and lack-of-density effect on MOR to RB. The product is produced by glue-laminating whole or half culms that have been softened under high heat and steam and progressively opened out flat through a series of custom-designed rollers [18,67]. Most engineered bamboo products have a low utilization rate of bamboo biomass and/or high content of adhesive, which can obscure the texture and grain of the original bamboo surface. Advantages of FB technology include its high recovery rate and maintenance of the original grain texture of bamboo without cracking, which makes it suited to appearance flooring and wall paneling applications. If the outer cortex is retained, then the utilization rate of flattened bamboo increases to around 90% [29]. Adhesive requirements are also greatly reduced compared with most other engineered bamboo products [68]. FB sheets are generally not heat-treated or heavily densified beyond about 1 g/cm^3^, giving them similar mechanical properties to RB, and can be glued into LBL-type products. Note that the MOR of RB, LBL, and FB ranges from 69.1–122.46 MPa and MOE from 3.6–17.9 GPa. The mechanical properties of these products are low and may only be suitable for lower structural performance requirements.

Bamboo scrimber (BS) is mostly denser and stronger than both the RB and LBL due to its higher densification ratio and required resin content. Note that BS and WBS have the highest variation in mechanical properties, with MOR ranging from 119–398 MPa and MOE from 13–32.3 GPa. Their strength properties are comparable to other structural building materials. From Figure 9, it can be seen that BLVL is notably less variable in properties than BS. Its mechanical properties are similar to BS but at a lower density. Other research shows BLVL is more uniform, allowing for better control and standardization properties [53]. BLVL is a relatively new product designed to improve the automation of the production process and the performance of the product. It also enhances material utilization to some extent. Its success lies in the use of stitched bamboo bundle mats to increase the laminate size and uniformity and hence achieve better bonding with less compaction requirement [36,69]. The greater uniformity of these elements effectively reduces variation in properties.

The width and extent of fiberization of BS strips also have a large impact on strength properties (as demonstrated here), as they affect both mat lay-up (see Figure 2) and resin content. BS currently lacks uniform manufacturing and quality standards that adequately account for the variation in density and mechanical properties of these products [70]. The use of wide strips and stitched veneer go a long way toward controlling axial alignment and the frequency of overlaps within layers, which requires greater compaction to achieve adequate surface contact and bonding than if there are no overlaps between layers, much like plywood or LVL [36].

Furthermore, the bamboo bundles used in BS often undergo a modification treatment such as carbonization prior to manufacture. Heat and steam treatments are used to improve the corrosion and mildew resistance, adjust product color, and improve dimensional stability. However, such treatments, particularly at higher temperatures, also contribute to variability in the mechanical properties of BS. It can be seen from Figure 9 that MOR and MOE of BS materials that had undergone saturated steam treatment (SST) and hot dry air treatment (HDAT) are in the range of 86–166.5 MPa and 4.43–29 GPa, respectively.

## 4. Conclusions

This study investigated the fiberization–density–property relationships of wide-bundle bamboo scrimber (WBS). The outer cortex of the culm was partially retained during fiberization, resulting in a utilization rate for WBS above 90% compared with 55% for normal scrimber. Fiberization creates cracks and exposes fiber surfaces in the bamboo strips. As such, the specific surface area (SSA) increased by 100%, and the resin absorption rate went up 2.7 times after four brooming passes. Upon consolidation and curing, the induced cracks were more-or-less “repaired”, forming a resin-infused mechanical interlocking network that enhanced water absorption, dimensional stability, and mechanical properties of the resulting composites. To balance product cost and performance, strips with 1–2 brooming passes and a panel density of 0.9–1.0 g/cm^3^ is recommended. This fiberization and density level yielded properties of MOR = 163.96–210.73 MPa, MOE = 16.11–20.9 GPa, and 28 h TS = 6.54–6.97%. These properties are, on average, better than regular BS, representing a significant improvement in product quality at lower compaction if wide strips (200–250 mm in width) are used. This was believed to be largely due to improved mat layering with better strip alignment and fewer random edge overlaps within layers compared with normal BS. The properties could be further enhanced by using strip stitching and continuous lay-up, leading to laminated structures such as bamboo laminated veneer lumber (BLVL).

The flexural properties (MOR and MOE) of WBS were compared with other engineered bamboo products. Those that use crack-free elements such as flattened bamboo (FB) or laminated bamboo lumber (LBL) have lower variations in density and flexural properties and are more similar to the bamboo parent tissue. While the MOE of LBL and FB increased with density, the MOR failed to show a significant increase with density, which is contrary to scrimber and counter-intuitive. This discrepancy can be attributed to the lack of sufficient bonding in the laminated products, causing premature failures in the glueline rather than inter-element stress transfer. The high densification of ‘fiberized’ element products such as bamboo scrimber or BLVL can improve mechanical properties due to high resin usage and mechanical interlocking. Further work will focus on the performance analysis of low-density and low-fiberization bamboo scrimbers to obtain a more accurate balance between material performance, bundle morphology, and product density.

## Figures and Tables

**Figure 1 materials-15-07518-f001:**
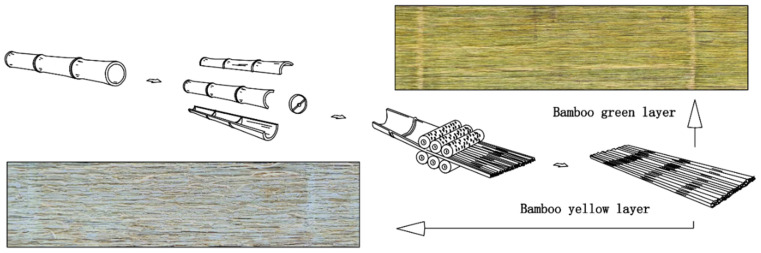
Culm flattening and fiberization process for wide bamboo scrimber (WBS). Image resolution = 300 dpi.

**Figure 2 materials-15-07518-f002:**
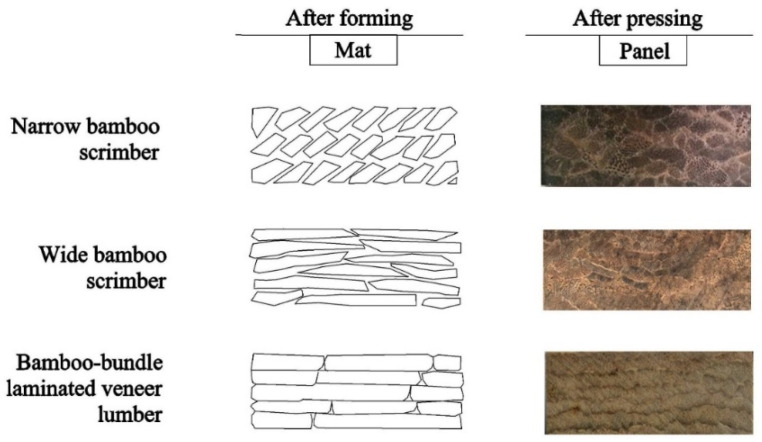
Cross-sections of mat lay-up and pressed panels for narrow BS, WBS, and bamboo bundle LVL.

**Figure 3 materials-15-07518-f003:**
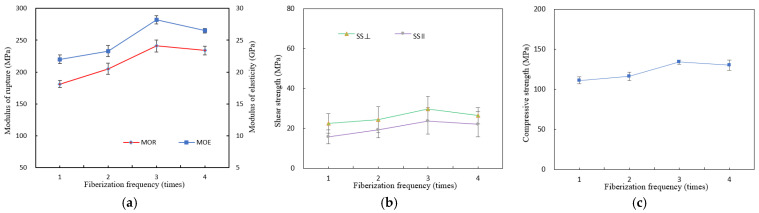
Mechanical properties of WBS with different fiberization frequency: (**a**) flexural modulus of rupture (MOR), modulus of elasticity (MOE), (**b**) parallel shearing strength (SS||), perpendicular shear strength (SS⊥), and (**c**) compressive strength (CS).

**Figure 4 materials-15-07518-f004:**
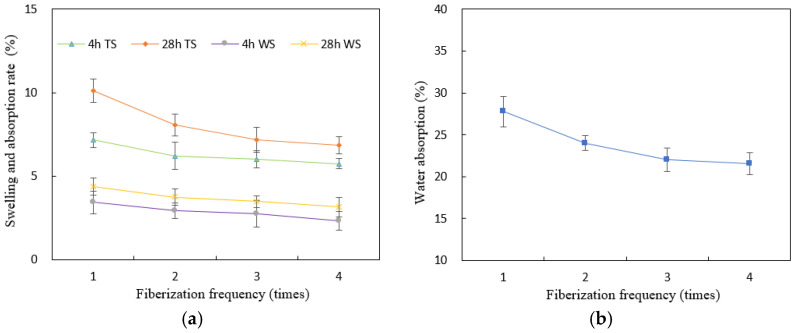
Dimensional stability at 4 h and 28 h of WBS with different fiberization frequency: (**a**) thickness swelling rate (TS), width swelling rate (WS), and (**b**) water absorption rate (WA).

**Figure 5 materials-15-07518-f005:**
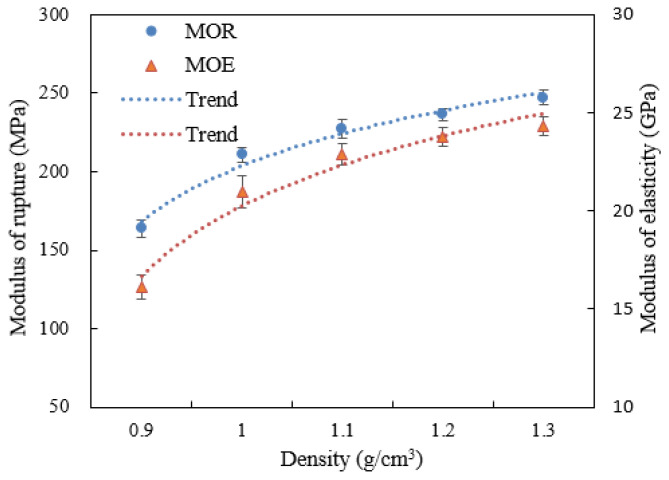
Effect of density on MOR and MOE of WBS.

**Figure 6 materials-15-07518-f006:**
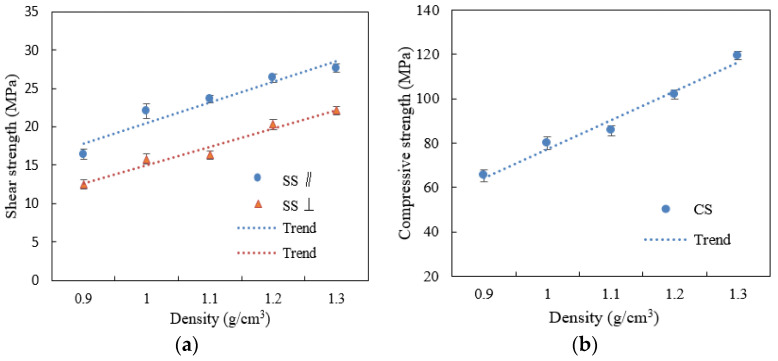
Effect of density on (**a**) shear strength (SS) and (**b**) compressive (CS) of WBS.

**Figure 7 materials-15-07518-f007:**
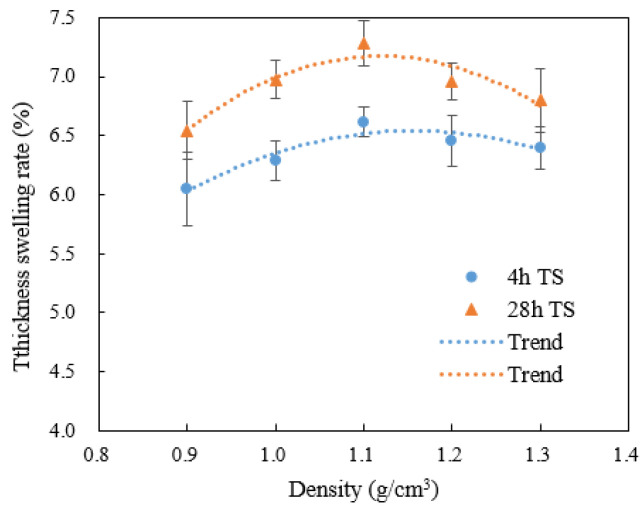
Effect of density on 4 h and 28 h TS of WBS.

**Figure 8 materials-15-07518-f008:**
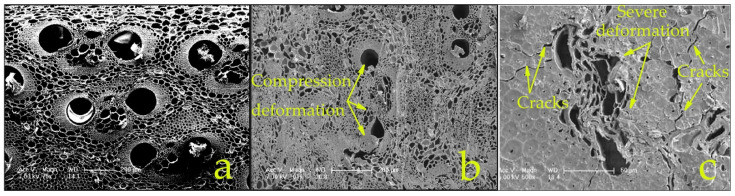
SEM images of raw bamboo and WBS: (**a**) raw bamboo density = 0.65 g/cm^3^, (**b**) WBS density = 0.9 g/cm^3^, (**c**) WBS density = 1.3 g/cm^3^.

**Figure 9 materials-15-07518-f009:**
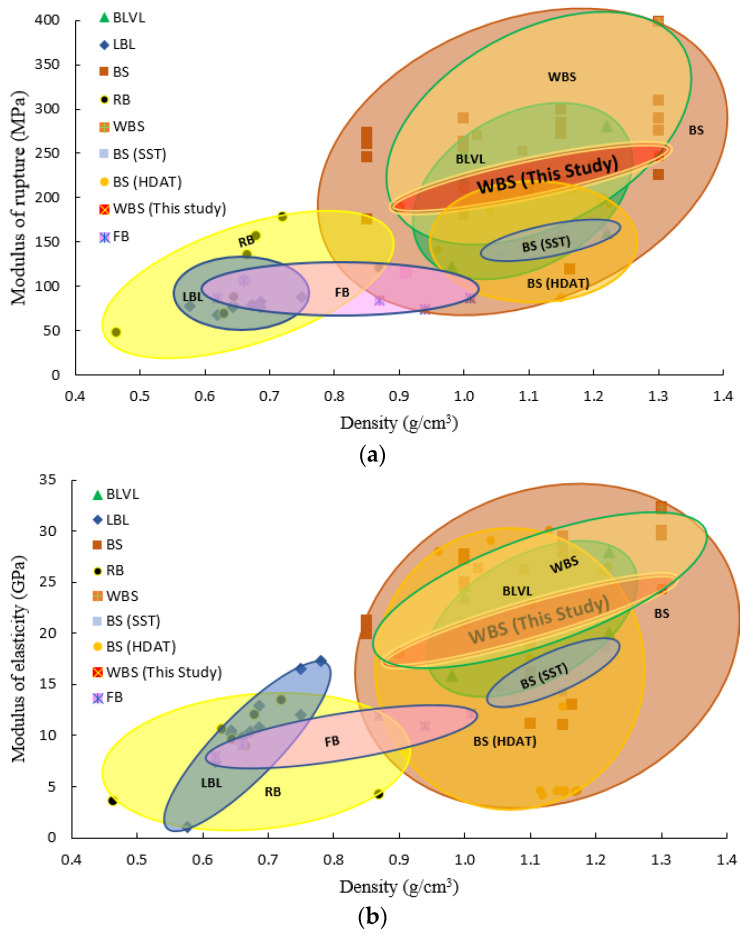
Density-flexural property relationships of bamboo products. (**a**) MOR, (**b**) MOE. Notes: BS—bamboo scrimber, WBS—wide bamboo scrimber, RB—raw bamboo, BLVL—bamboo bundle laminated veneer lumber, LBL—laminated bamboo lumber, and FB—flattened bamboo. All bamboo composites use Moso (*Phyllostachys pubescens* Mazel.) and RB includes Moso, Guadua and Dendrocalamus asper.

**Table 1 materials-15-07518-t001:** Experiment design for fiberization frequency and density sampling.

Expt.	Experimental Factors	Level				
a	Board density (g/cm^3^)	1.00				
	Fiberization frequency (times)		1	2	3	4
b	Fiberization frequency (times)	3				
	Board density (g/cm^3^)	0.90	1.00	1.10	1.20	1.30

**Table 2 materials-15-07518-t002:** Morphological characteristics of WBS, NBS, and traditional Moso bamboo bundles.

Bundle Type	Number of Strips per Culm	Width (mm)	Thickness (mm)	Diameter (mm)	Utilization Rate (%)	Photo of Bundles	
Wide-bundle Bamboo Scrimber	3–4	200–250 (234)	10–12 (10.95)	1.35–1.72 (1.54)	92	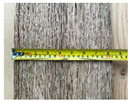	
Narrow-bundle Bamboo Scrimber	4–8	30–60 (45)	10–12 (10.32)	1.15–1.52 (1.34)	80	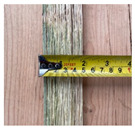	
Traditional Bamboo Scrimber bundles	6–10	20–30 (23)	8–10 (8.32)	2–3 (2.32)	50–55	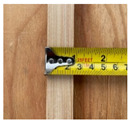	

The numbers in parentheses are average values. Image resolution = 300 dpi. The data in the figure are summarized from experimental data from the Wood-based Panel and Adhesive Laboratory of the Chinese Academy of Forestry and the laboratory of the Institute of Forest Industry, Jiangxi Academy of Forestry.

**Table 3 materials-15-07518-t003:** Morphological characteristics of WBS bundles with different fiberization frequency.

Fiberization Frequency (Times)	Outer Green Removal Rate (%)	Diameter of Bamboo Bundles (mm)	Resin Loading (%)	Specific Surface Area (1/mm)	Cross-Sectional Photo of Bamboo Fiber Bundles (5 mm Scale Bar)
1	34.23 (4.75)	4.21 (0.32)	26.45 (4.32)	1.74	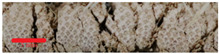
2	56.76 (8.37)	3.13 (0.21)	47.21 (5.13)	2.41	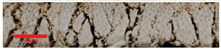
3	79.11 (10.88)	2.54 (0.13)	58.17 (4.55)	3.32	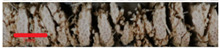
4	94.23 (13.23)	1.89 (0.15)	72.44 (6.73)	3.54	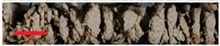

Numbers in parentheses are coefficients of variation. Data summarized from experiments at the Wood-based Panel and Adhesive Laboratory of the Chinese Academy of Forestry and the laboratory of the Institute of Forest Industry, Jiangxi Academy of Forestry.

**Table 4 materials-15-07518-t004:** The mechanical properties of different engineered bamboo products.

Products	Density (kg/cm^3^)	Modulus of Rupture (MPa)	Modulus of Elasticity (GPa)	Reference	
WBS	1.02–1.3	253.23–398	26.35–32.3	Yu et al. [30]	
				Yu et al. [43]	
BS	0.85–1.3	119–398	13–32.3	Huang et al. [31]	Yu et al. [15]
				Sharma et al. [44]	Shang et al. [27]
				Wei et al. [45]	Zhang et al. [46]
				Yu et al. [17]	Kumar et al. [47]
RB	0.68–0.87	69.1–122.46	3.6–13.5	Lorenzo et al. [48]	Yu et al. [17]
				Ribeiro et al. [49]	Sharma et al. [44]
				Chung et al. [50]	Huang et al. [51]
				Dixon et al. [52]	Yu et al. [30]
BLVL	0.98–1	121.31–280	15.83–24.61	Zhou et al. [53]	He et al. [54]
				Chen et al. [55]	Deng et al. [56]
				Deng et al. [57]	
LBL	0.62–0.78	67.7–88	7.4–17.3	Andy et al. [58]	Yu et al. [30]
				Rittironk et al. [59]	Sharma et al. [60]
FB	0.62–1.01	74–115.13	4.49–12.1	Andy et al. [58]	Huang et al. [31]
				Rittironk et al. [59]	Sharma et al. [44]
				Wenji Yu et al. [30]	Nugroho et al. [61]

## Data Availability

No new data were created or analyzed in this study. Data sharing is not applicable to this article.

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
