# Peer review of "Manufacturing and Characterization of Wide-Bundle Bamboo Scrimber: A Comparison with Other Engineered Bamboo Composites"

_materials, 2022, doi:10.3390/ma15217518_

Round 1
Reviewer 1 Report
This article is important to explain the utilization of bamboo material which is one of renewable resources as fast-growing plant. Wide-bundle bamboo scrimber board has been compared with many engineered bamboo board including traditional board. I think that the readers read it with interesting. There were some questions as following, but I would like to decide to accept it if you can revise it.
Line 159-161:
Could you show the resolution of image (for example, mm/pixel)?
Line 217-218, Figure 2:
Could you show the scale of image or picture (for example, mm/pixel)?
Line 232-236:
Figure 3 explains that three of four times in fiberization frequency result in the miximum MOE and MOR. In this article, however, two or three times in fiberization is more effective then four times to consideration of energy consumption and cost. I can not understand it because there are no explanation of realistic energy consumption value and cost. Could you refer any previous study if you have.
Line 387-389, Figure 9.
Ashbly plots in Fig.9 is so interesting. Are almost of the properties in MOR and MOE for Boso Bamboo? Could you show the species of the bamboo material if possible?
Author Response
Thank you for the review. Please see the responses to each line item in the attachment.

Reviewer 2 Report
This manuscript aims to examine the effects of fibrisation frequency and board density on the mechanical properties of wide bundle scrimber panels. The motivation of the research and experimental design are simple and clear, and the result itself is also reasonable. The result would attract the readers who are involved in the wood based panel/beam production. However, there are some orthographic variants and comprex figures. I think this manuscript needs some revision. L139-140 "Several pieces of fiberized bamboo veneer were randomly selected and the diameter of the thickest bundle of fibers in each piece was measured using vernier calipers" Cross section of the fiber does not like a round shape. How did authors define the diameter? L141-142 Table 3 -> Table 2 L181-191 Please clarify the size of the specimens and number of replications for each test. Figure 3 Figure 3 should be separated by at least 3 figures; a) MOR and MOE, b) shear strength, c) compressive strength. Figure 3, L258 There are some orthographic variants; shearing/shear strength, compressive/compression strength. Please unify them. Figure 3, L258-261 "Compression strength (CS) and shear strength (SS) showed a steady increase with the number of passes as this is governed by the interface strength between two laminated faces. More passes mean more fissures in the adherend surfaces and more resin in the panel, creating stronger and better resin interlocking between adherence and stronger bond interfaces." According to Figure 3, the trend of both SS and CS is similar to flexural properties. They increased by three passes but there is no obvious increase at four passes. Figure 4 It would be better to separate the figure into 2; a)TS and WS, b) WA. Figure 4, L263-264 Dimensional stability indicators thickness swell (TS), width swell (WS) and water absorption (WA) significantly reduced with fiberization frequency from one to three with a little further gain after four passes. Did authors verify the significant reduction by the statistical test? I recommend authors to conduct multiple comparison tests. Figure 5 Equation of the trents and R square would help us to understand the relationship between density and flexural properties. Figure 8 It is more informative if authors have those different compression images with the same magnification. The Yellow colored scale bar on Fig.8(b) might be wrong. Original scale bar is shown on the right side of the yellow one, but the original one is clearly short.Author Response
Thank you for the review. Please see the responses to each line item in the attachment.

Reviewer 3 Report
Dear Authors,
The manuscript is well prepared. I have only a few remarks found in the attached file.
Please take care especially to the abbreviations; there are quite many of them, all need to be specified and properly used in the text.
Best regards

Author Response

(The authors gave the same response as above.)
